# Recovery of ΔF508-CFTR Function by Citrate

**DOI:** 10.3390/nu14204283

**Published:** 2022-10-14

**Authors:** Beatrice Borkenhagen, Peter Prehm

**Affiliations:** 1Institute of Physiological Chemistry and Pathobiochemistry, Muenster University Hospital, Waldeyerstr. 15, 48129 Muenster, Germany; 2Institute for Anatomy and Experimental Morphology, University Hospital Hamburg-Eppendorf, Martinistrasse 52, 20246 Hamburg, Germany

**Keywords:** cystic fibrosis, CFTR, citrate, hyaluronan

## Abstract

Treatment of cystic fibrosis relies so far on expensive and sophisticated drugs. A logical approach to rescuing the defective ΔF508-CFTR protein has not yet been published. Therefore, virtual docking of ATP and CFTR activators to the open conformation of the CFTR protein was performed. A new ATP binding site outside of the two known locations was identified. It was located in the cleft between the nucleotide binding domains NBD1 and NBD2 and comprised six basic amino acids in close proximity. Citrate and isocitrate were also bound to this site. Citrate was evaluated for its action on epithelial cells with intact CFTR and defective ΔF508-CFTR. It activated hyaluronan export from human breast carcinoma cells and iodide efflux, and recovered ΔF508-CFTR from premature intracellular degradation. In conclusion, citrate is an activator for ΔF508-CFTR and increases export by defective ΔF508-CFTR into the extracellular matrix of epithelial cells.

## 1. Introduction

Cystic fibrosis (CF) is one of the most common inherited diseases, afflicting 1 in approximately 2500 white individuals [1]. The primary cause of morbidity and mortality in CF is chronic lung infection and deterioration of lung function. CF is caused by mutations in the CF transmembrane conductance regulator (CFTR) gene, which encodes a chloride channel expressed at the apical membrane of epithelial cells in the airways, pancreas, testis, and other tissues [2,3]. The most common CFTR mutation producing CF is deletion of phenylalanine at residue 508 (ΔF508) in its amino acid sequence. The ΔF508-CFTR protein is misfolded and retained at the endoplasmic reticulum, where it is degraded rapidly. Small-molecule activators of defective ΔF508-CFTR folding/cellular processing (“correctors”) and channel gating (“potentiators”) may provide a strategy for therapy of CF that corrects the underlying defect. A number of small-molecule ΔF508-CFTR potentiators and correctors have been identified [4]. They were found by high-throughput screening for activation of the chloride channel.

We discovered that hyaluronan is exported from mesenchymal fibroblasts by MRP5 (multidrug resistance associated protein 5) [5] and from epithelial cells by CFTR [6]. Hyaluronan has an important role in epithelial clearing of the bronchial surface by facilitating cilial beating [7,8]. In an attempt to evaluate interaction between CFTR and hyaluronan transmembrane transport, we synthesized a new class of drug-like compounds (Hylout4) that mimic the non-reducing end of hyaluronan and discovered that they activated chloride export from bronchial epithelial cells and hyaluronan export from breast cancer cells [9]. The compounds together with other known modulators were docked to the open and closed conformation of ΔF508-CFTR [10] to identify possible optimized activators for ΔF508-CFTR. Molecular modelling and physiological experiments showed that citrate serves as an activator of hyaluronan transport in ΔF508-CFTR.

## 2. Materials and Methods

### 2.1. Materials

Mouse-anti-CFTR-IgM was from Acris Antibodies, Hiddenhausen, Germany. Other chemicals were from Sigma-Aldrich Chemical Corporation (St. Louis, MO, USA).

### 2.2. Computational Studies

In silico docking studies were carried out to evaluate the affinity and binding interactions by molecular docking simulation using the AutoDock 4.2 and PyRx software—Python Prescription 0.8 to coordinates of the open and closed CFTR conformation [10,11,12]. ATP, citrate and docosahexaenoic acid (DHA) were docked as fully ionized molecules to enable ionic interaction with basic amino acids. The theoretical K_I_-values of binding are listed in Table 1.

### 2.3. Cell Lines

The cell line HMT3552 has been described previously [18]. The wildtype CFTR (16HBE14o^−^) and the mutant ΔF508-CFTR (CFBE41o^−^) cell line were from Dr. D.C. Gruenert [19].

### 2.4. Western Blotting of CFTR

Western blotting was performed as described [9].

### 2.5. Determination of Hyaluronan Export

Hyaluronan export of HMT3552 cells was performed as described [20].

### 2.6. Iodide Efflux

Iodide efflux experiments were performed as described [21].

## 3. Results

### 3.1. Virtual Docking

In an attempt to identify optimized CFTR activators, we docked Hylout4 [9] and several other known activators and inhibitors to the open and closed conformation of CFTR [10,12]. In preliminary docking calculations, we found that the outward-facing CFTR configuration [10,11] bound Hylout4 with higher affinities than the inward-facing ATP free configurations. The predicted Hylout4 binding site was a cleft between the nucleotide binding domains NBD1 (433 to 634) and NBD2 (1225 to 1415). The CFTR activators corr-4a, CFTR_act_-06, VX809, the inhibitor CFTR_inh_-172 and dietary supplement docosahexaenoic acid (DHA) also bound to this cavity with the highest affinity (Table 1). It comprised the amino acids R170, A171, E267, W465, F494, R553, K968, K1060, W1063, K1292, G1342, and K1351. The most striking feature of this site was the close neighbourhood of six basic amino acids (underlined) and the preference of lysines that have long flexible arms for ligand binding by ionic interactions and hydrogen bonds. The distances of positive charges in the open conformation ranged from 5 Å to 16 Å and in the closed conformation from 12 Å to 32 Å.

Evaluating other potential binding compounds, we included substrates of the intermediate metabolism and found that citrate and isocitrate showed considerable affinity for this binding site (Table 1, Figure 1).

Figure 1a shows the location of citrate in the cleft between the nucleotide binding domains. Figure 1b shows that citrate and the phosphate groups of ATP are located at the same position in close vicinity of F508, which was 11 Å apart from R553.

### 3.2. Activation of Hyaluronan Export

Since hyaluronan is exported from human breast carcinoma cells by CFTR [5], citrate and isocitrate were tested for their influence on hyaluronan export in cell culture. Figure 2 shows that both compounds activated hyaluronan export in a concentration-dependent manner. Citrate showed a higher increase of hyaluronan export than isocitrate. Therefore, citrate was chosen for further evaluation.

### 3.3. Iodide Efflux

In addition to chloride and hyaluronan, CFTR also exports iodide that can be measured by an iodide selective electrode. Iodide efflux from 16HBE14o- and CFBE41o- cells was assessed with increasing citrate concentrations. Citrate activated immediately iodide efflux from 16HBE14o- as well as from CFBE41o- cells (Figure 3), indicating that citrate recovered the activity of ΔF508-CFTR.

### 3.4. Recovery of ΔF508-CFTR Cell Surface Expression

The ΔF508-CFTR mutation impairs maturation and destabilizes the protein in post-Golgi compartments. Figure 4 shows that citrate increased ΔF508-CFTR cell surface expression, indicating that citrate recovered cellular processing of ΔF508-CFTR.

## 4. Discussion

It is surprising that virtual docking of ATP has so far been conducted on computed CFTR fragments [22], but not the whole structural model, in spite of the long history of intensive pharmaceutical research. It is even more surprising that the highest docking affinity for ATP and other activators of CFTR function did not reside in the two known ATP binding sites, but in a cleft between the nucleotide binding sites NBD1 and NBD2. This site has been claimed to open the CFTR channel due to salt bridges between opposing basic and acidic residues [23]. Remarkably, the external phosphate of ATP overlaps with the positions of citrate and the carboxyl group of DHA. Thus, our docking results confirm the notion of salt bridges between the NBD1 and NBD2 domains; however, we propose that the salt bridges are mediated by anionic activators.

These basic amino acids are probably responsible for ionic nucleotide binding and stabilization of the open CFTR conformation. The ATP binding affinity to this binding site is very high with K_D_ = 22 nM in comparison to normal intracellular nucleotide triphosphate concentrations in the mM range, but such high ATP concentrations are required for reliable gating [24,25]. The ATP binding sites had previously been located by site-directed CFTR mutations to W401 in NBD1 and Y1219 in NBD2 [26] which are only 9.8 Å and 11.4 Å apart from our predicted amino acids R170 and K968, respectively. Virtual docking also revealed that the most known CFTR activators bound to this cavity by bridging both nucleotide binding domains and thus appear to function as ATP agonists stabilizing the open conformation. This mechanism of action has been shown for the CFTR activator VX-770, which opens the ΔF508-CFTR in an ATP-independent manner [27] and decouples gating from ATP hydrolysis [28]. It is interesting to note that also the inhibitor CFTR_inh_-172 had high affinity to the ATP binding site. It could be an ATP antagonist which displaces ATP, destabilizing the open conformation.

The conspicuous proximity of six basic amino acids led us to investigate other possible ligands. We tested citrate and isocitrate and found considerable affinity of K_D_ = 2.7 µM and K_D_ = 1.1 µM, respectively. These theoretical affinities were again much higher than the normal cytosolic concentrations of 80 µM to 220 µM for citrate and about 10 µM for isocitrate [29,30], indicating that high saturation density is again necessary for gating. The levels of both ATP and citrate are indicators of the cellular energy level [31].

Accordingly, our cell culture experiments showed that citrate activated hyaluronan export from breast carcinoma cells, stimulated iodide efflux from epithelial cells carrying normal CFTR as well as ΔF508-CFTR and recovered it from premature intracellular degradation. It can thus be regarded as a potentiator as well as a corrector for the defective ΔF508-CFTR protein.

Part of the cytosolic citrate is exported into the circulation, giving rise to citrate in plasma and urine. It is known that urinary citrate secretion from cystic fibrosis patients was significantly lower [32,33]. A striking similarity of citrate and hyaluronan is their high Ca^2+^-chelating property with K_D_ = 171 µM [34] and K_D_ = 1.4 mM [35], respectively, as compared to the normal plasma concentration of about 2 mM. Cystic fibrosis patients have abnormal Ca^2+^ homeostasis [36], and chelate formation of cytosolic citrate and Ca^2+^ could reduce the effective Ca^2+^ concentration which is required for normal ΔF508-CFTR trafficking [37].

The function of citrate as Ca^2+^ chelator and as hyaluronan export activator may explain some of the observed metabolic disturbances of cystic fibrosis patients such as the associated diabetes [38] and lithiasis [33]. Diabetes favours a shift from the aerobic energy production by the citric acid cycle and respiration towards gluconeogenesis, causing withdrawal of metabolites from the citric acid cycle. Low citrate in other tissues such as kidney again decreases extracellular hyaluronan required for Ca^2+^ complexation and solubilization, causing lithiasis in cystic fibrosis patients [39,40,41,42]. Patients thus remain in a vicious cycle, unless they are treated by citrate as a food additive for stone solubilization [43,44,45,46,47,48,49,50]. Citrate is freely available in the supermarket and also a nutritional additive for treatment of lithiasis.

Citrate has been patented as an ingredient of inhalation solutions to reduce the viscosity of the sticky mucous in cystic fibrosis patients [51]. It was speculated that citrate might correct all other symptoms of cystic fibrosis by systemically recovering the function of ΔF508-CFTR.

Citrate and isocitrate are intermediates of the citric acid cycle within mitochondria. They are nutrients taken up by mammalian cells and form two separated citrate pools: mitochondrial and cytosolic. The intracellular citrate level depends on nutrition: A carbohydrate-rich nutrition induces the citrate lyase to reduce the citrate concentration by transformation into fatty acids. A protein-rich nutrition induces anaplerotic reactions to increase intracellular citrate concentrations. A protein-rich diet has frequently been evaluated over the past decades with no final conclusion, but it seemed to be underestimated [52] probably because activation of hyaluronan export by citrate by the regulatory balance of carbohydrate-rich and protein-rich diets was not taken into account. Indeed, a low glycemic index diet has already been shown to be beneficial for cystic fibrosis patients [53,54].

Frequent and severe infections of cystic fibrosis patients by *Pseudomonas aeruginosa* are often responsible for morbidity in CF patients. These bacteria can upregulate an isocitrate lyase as a virulence factor for persistent infection [55], which further depletes citrate levels and thus inactivates hyaluronan export. Sufficient hyaluronan shedding is required for efficient ciliary beating and debris removal from airways [8]. Thus, a protein-rich and low carbohydrate diet supplemented with DHA and citrate could have benefits similar to expensive drugs and lucrative treatments.

## Figures and Tables

**Figure 1 nutrients-14-04283-f001:**
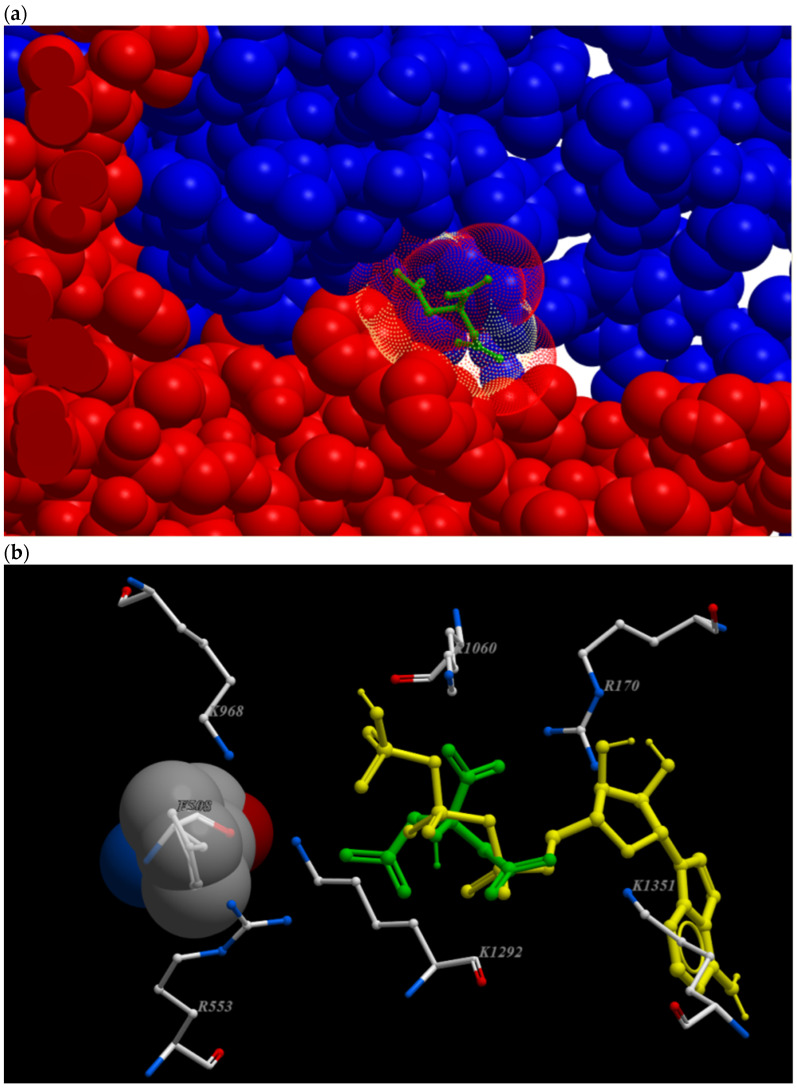
Binding site of citrate and ATP to the open conformation of CFTR. (**a**) Citrate (green) binds to a cleft between the nucleotide binding domains NBD1 (red) and NBD2 (blue). (**b**) Citrate (green) and ATP (yellow) bind to basic amino acid residues in the vicinity of F508.

**Figure 2 nutrients-14-04283-f002:**
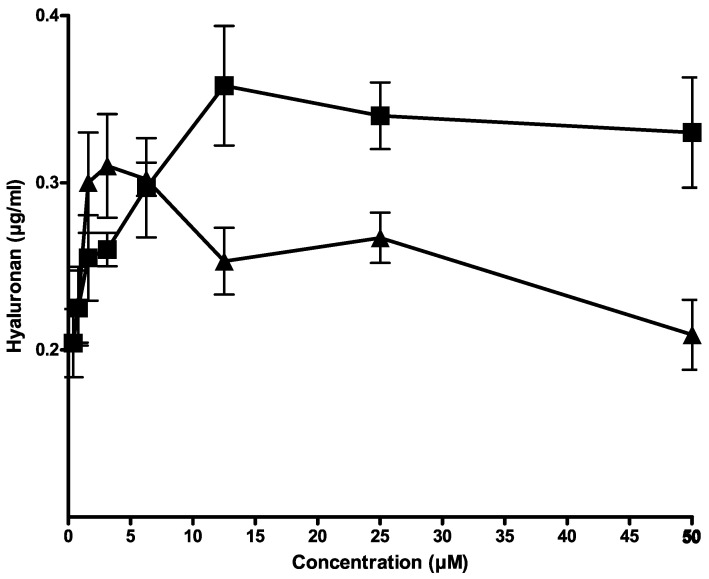
Activation of hyaluronan export by citrate (■) and isocitrate (▲) from HMT3552 cells. The error bars indicate the means of two determinations.

**Figure 3 nutrients-14-04283-f003:**
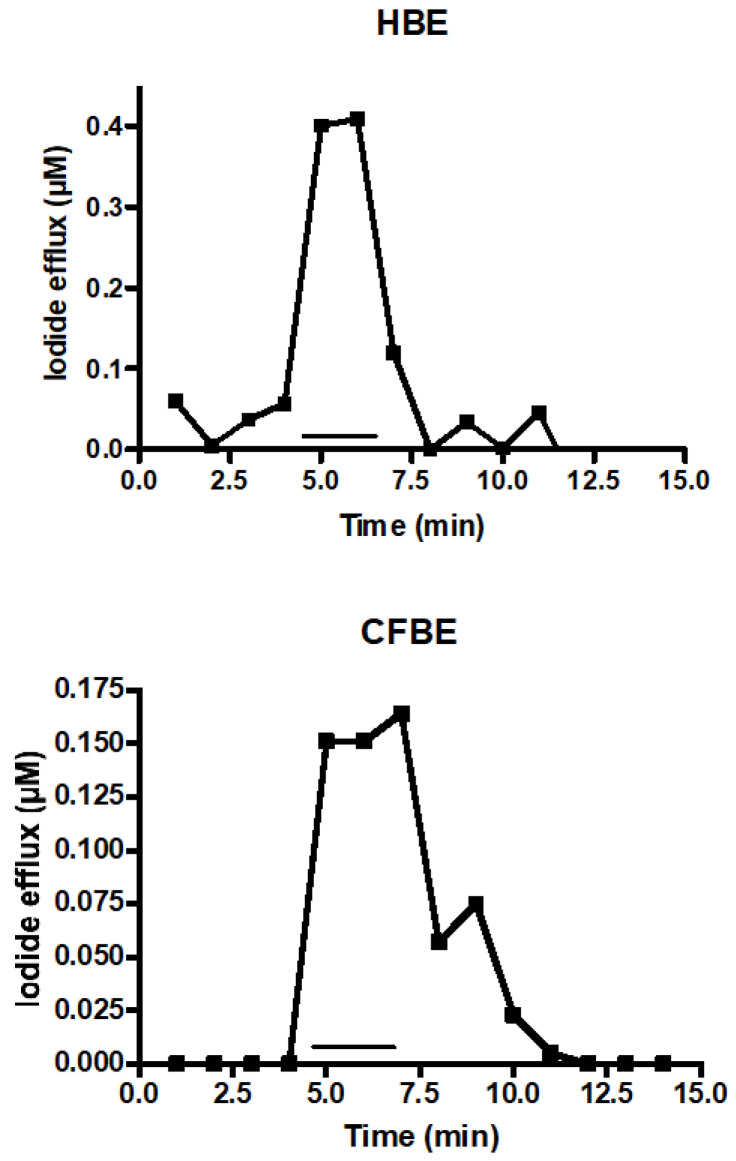
Citrate stimulated iodide efflux. CFTR can also export iodide instead of chloride. 16HBE14o- or CFBE41o- were loaded with iodide and the iodide concentration was measured by an iodide-sensitive electrode in 1 min intervals. The bar indicates the time period of exposure to 100 µM of citrate.

**Figure 4 nutrients-14-04283-f004:**
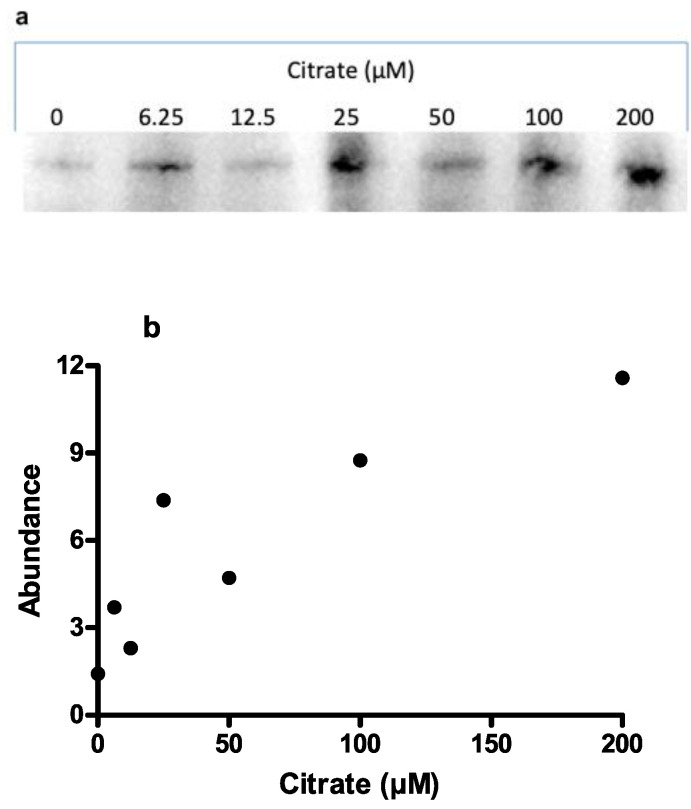
(**a**) Citrate recovered from ΔF508-CFTR cell surface expression as analysed by Western blotting. (**b**) relative abundance of cell surface-expressed ΔF508-CFTR.

**Table 1 nutrients-14-04283-t001:** Virtual docking of activators and inhibitors to the open conformation of CFTR.

Name	Structure	Calculated Affinity Km (µM)	Experimental Activation/InhibitionEC50 (µM)	Ref.
Hylout4	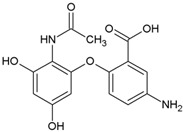	7	100for HA	[9]
ATP	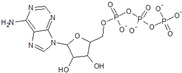	0.022	41for Cl^−^	[13]
CFTRact-06	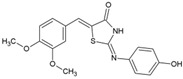	1.4	1for HA	[6]
Corr-4a	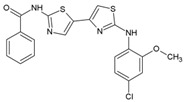	0.01	1for Cl^−^	[14]
VX809	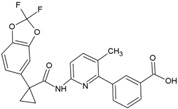	0.01	0.081for Cl^−^	[15]
CFTR_inh_127	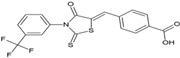	0.22	0.3for Cl^−^	[16]
DHA	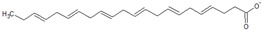	17		[17]
Citrate	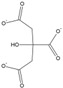	2.7	6for HA	
Isocitrate	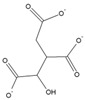	1.1		

Docking was performed with the AutoDock software 1.5.6.rc3. The affinities of the best fits are listed.

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
