# Peer review of "Recovery of ΔF508-CFTR Function by Citrate"

_nutrients, 2022, doi:10.3390/nu14204283_

Round 1

Reviewer 1 Report

The communication presents the results on the ability of citrate molecule to activate mutated CF transmembrane conductance regulator (CFTR) protein. Authors used docking studies to predict binding site of the potential activators of F508-CFTR including citrate and isocitrate. The results are well presented and provide interesting findings. The paper is concise and well discussed. Thera era only small issues that need to be addressed before publication.

Line 31 Please provide the full meaning of the abbreviation MRP5

If table 1 stays in that position I recommend ordering the references in the table accordingly  

Line 95 please rephrase the sentence “Citrate showed….” It is unclear what is the “constant increase” and “activating a lower concentrations” referring to.

Figure 2. The legend is missing for the two lines on the graph (citrate and isocitrate)

I would recommend adding some newer references. Among total of 54 references, only 4 are published after 2014. i.e. in the last 10 years.

Author Response

We added the meaning of MRP5:

We discovered that hyaluronan is exported from  mesenchymal fibroblasts by MRP5 (multidrug resistance associated protein 5) ...

We reordered the citations in Table 1.

We rephrased the sentence in the following way:

Citrate showed a higher  increase of hyaluronan export than isocitrate. Therefore, citrate was chosen for further evaluation.

Figure 2: we added the symbols for citrate and isocitrate:

Fig. 2. Activation of hyaluronan export by citrate (■)  and isocitrate (▲) from HMT3552 cells. The error bars indicate the means of two determinations.

We added two more references published after 2014:

Indeed, low glycemic index diet has already been shown to beneficial for cystic fibrosis patients [54,55].

Reviewer 2 Report

The study by Borkenhagen and Prehm describes results obtained in silico and in vitro, showing that citrate is an activator of hyaluronan transport in F508-CFTR. While the docking approach is solid, a few cell-based experiments lack clarity.

Some comments to improve the manuscript:

The abstract is too brief and difficult to interpret at first read. One sentence explaining the background and the study question would help.

Abbreviations should be followed by the full name the first time they are mentioned, eg MRP5, line 32.

Why is hyaluronan-CFTR important physiologically? Why is hyaluronan exported from cancer cells? The introduction should cover aspects of hyaluronan involvement in cystic fibrosis.

Table 1: Check the units' decimal representation; replace the comma with dots (columns Km and EC50).

Figure 2 legend: the labels of the experimental dots (squares, triangles for citrate, isocitrate) are missing. 

Figure 4: the blots should be complemented with a bar graph showing the density of the bands to analyze the differences better. 

Lines 158-160: the authors describe effects that were not assessed in this study. Either remove or insert references.

The discussion is quite broad, it should be more focused on discussing the results.

Author Response

We expanded the short abstract by the following sentences:

Treatment of cystic fibrosis relies so far on expensive and sophisticated drugs. A logical approach to rescuing the defective △F508-CFTR protein has not yet been published. Therefore virtual docking of ATP and  CFTR activators to the open conformation of CFTR protein was performed. A new ATP binding site outside of the two known locations was  identified.

We added the full name of MRP5:

We discovered that hyaluronan is exported from  mesenchymal fibroblasts by MRP5 (multidrug resistance associated protein 5) ...

Why is hyaluronan-CFTR important physiologically?

The importance of hyaluronan for cystic fibrosis was addressed by adding the following phrase in the introduction:

Hyaluronan has an important role in epithelial clearing of the bronchial surface by facilitating cilial beating [7,8].

Why is hyaluronan exported from cancer cells?

Hyaluronan synthesis from cancer cells is currently a broad field of extensive research with no definitive conclusions. We used this cell line here as an epithelial cell line which contains CFTR and produces hyaluronan only to access to effect of citrate on hyaluronan export. We believe that further discussion of hyaluronan from cancer cells will not be relevant in this paper.

Figure 4: We added a bar graph for the density of the bands.

Table 1: the commas were replaced with dots.

We removed the sentence in the lines 158 to 160.

The Discussion indeed appears to be quite broad and the first reviewer even asked for more citations published after 2014. So we followed his advice and added to more references.

On the other hand, we found it necessary to discuss not only our results, but also their implications for potential clinical trials or treatments in view of the central role of citrate in intermediate metabolism, because it not only connected to glucose metabolism and calcium homeostasis, but also to hyaluronan export.

Round 2

Reviewer 2 Report

The authors addressed all the raised comments and improved the manuscript.